# Rearing Undocked Pigs on Fully Slatted Floors Using Multiple Types and Variations of Enrichment

**DOI:** 10.3390/ani9040139

**Published:** 2019-04-02

**Authors:** Jen-Yun Chou, Constance M. V. Drique, Dale A. Sandercock, Rick B. D’Eath, Keelin O’Driscoll

**Affiliations:** 1Pig Development Department, Teagasc, P61 P302 Moorepark, Ireland; keelin.odriscoll@teagasc.ie; 2Animal & Veterinary Sciences Research Group, SRUC, Roslin Institute Building, Easter Bush, Midlothian EH25 9RG, UK; dale.sandercock@sruc.ac.uk (D.A.S.); rick.death@sruc.ac.uk (R.B.D.); 3Royal (Dick) School of Veterinary Studies, University of Edinburgh, Easter Bush, Midlothian EH25 9RG, UK; 4Agrocampus Ouest, 35042 Rennes, France; constance.drique@agrocampus-ouest.fr

**Keywords:** pig, environmental enrichment, slatted system, tail biting

## Abstract

**Simple Summary:**

Floors with a series of gaps to allow pig manure to pass through (slatted floors) are common in the pig industry as they enable the efficient management of pig waste. Pigs need enrichment materials to occupy them and to reduce harmful behaviours like tail biting. Loose materials are problematic in slatted systems because they can block up the slats and slurry pumps. This study aimed to establish if it is possible to rear pigs with undocked tails in a fully slatted system with compatible enrichment, while keeping tail biting at a manageable level and to investigate how important the variation of enrichment is. The results showed that although some tail biting occurred, the level was low with only mild lesions observed. Only 1 out of 96 pigs sustained severe tail damage (portion of tail bitten off). Pigs receiving a more varied enrichment tended to have lower tail lesion scores than pigs continuously presented with the same enrichment. This study showed that an optimal quantity and quality of slat-compatible enrichment provision can reduce the risk of tail biting in undocked pigs in fully slatted systems. The important role of adequate environmental enrichment provision in fulfilling pigs’ biological needs cannot be overemphasised.

**Abstract:**

In fully slatted systems, tail biting is difficult to manage when pigs’ tails are not docked because loose enrichment material can obstruct slurry systems. This pilot study sought to determine: a) whether intact-tailed pigs can be reared with a manageable level of tail biting by using multiple slat-compatible enrichment; b) whether a variation of enrichment has an effect; and c) whether pigs show a preference in enrichment use. Ninety-six undocked pigs were given the same enrichment items from one week after birth until weaning. At weaning, four different combinations of 8 enrichment items were utilized based on predefined characteristics. These were randomly assigned to 8 pens (*n* = 12 pigs/pen). Four pens had the same combination (SAME) from assignment and four pens switched combinations every two weeks (SWITCH). Individual lesion scores, interactions with the enrichment, and harmful behaviours were recorded. The average tail score during the experiment was low (0.93 ± 0.02). Only one pig in a SAME pen had a severely bitten tail (partly amputated). The overall level of interaction with enrichment did not decline over time. Pigs interacted with a rack of loose material most frequently (*p* < 0.001). The study showed promising results for rearing undocked pigs on fully slatted floors using slat-compatible enrichment.

## 1. Introduction

Tail biting is a damaging behaviour performed by pigs which causes injuries and pain in the recipients, with the worst cases causing permanent spinal infection and death [1]. The economic loss resulting from premature death and carcass condemnation due to tail biting is a serious issue for the pig industry [2,3]. Therefore, in order to control tail biting, tail docking is often practiced; there is a lower tail biting prevalence in docked than in undocked pigs [4]. However, routine tail docking is not permitted in the EU [5], and it can only be performed if other measures have been shown to be unsuccessful in controlling tail biting. It is, therefore, important to both encourage changes in attitudes and practices in the pig industry to ensure tail docking will be phased out and to provide practical and feasible advice to farmers on how to manage tail biting in different production systems.

Tail biting is most commonly observed when pigs are housed in suboptimal conditions (e.g., a high stocking density [6], high levels of competition for feed or water, and poor ventilation [7]). A lack of environmental stimulation for pigs has also been recognised as a key risk factor for tail biting [7]. In countries where tail docking is completely banned, loose materials are usually provided on the solid floor in fully concrete or part-slatted pens [8]. Fully slatted floors are, thus, considered a major risk factor for tail biting damage [9,10], at least in part because substrates are not usually placed on the floor as it could obstruct slurry storage systems [11]. Different organic enrichment materials that are not presented as litter and, therefore, more compatible with slatted systems have been investigated, mainly with docked pigs [12,13,14,15], but there has been limited evidence of efficacy where tail biting risk is high. Haigh et al. found that compressed straw blocks were no more effective than hanging rubber toys at reducing levels of damaging behaviour or tail lesion scores in docked pigs [14]. Likewise, other studies found no benefits to wood, when compared with a rubber floor toy [16]. Holling et al. also reported inconclusive effects on tail biting when using a straw dispenser due to its low occurrence [15], and Bulens et al. did not record any tail lesion when comparing different straw dispensers in docked pigs [12]. In undocked pigs, Veit et al. provided pigs with loose material (either corn silage or alfalfa hay) from the second week of life until 40 days post weaning and found that, although corn silage sustained pigs’ interest and reduced tail lesion prevalence, the percentage of pigs with some tail loss was over 50% in all treatments by the end of the experiment [17]. Zonderland et al. showed that a small amount of long straw in a rack could reduce the presence of tail wounds although still not as effective as an even smaller amount of loose straw presented on the floor [18]. There was also an issue with the blockage of the manure system when using long straw in the straw rack.

The enrichment materials which pigs prefer to use are characterised by being destructible, odorous, chewable, deformable, and ingestible [19,20]. A common perception exists that enrichment devices or materials will be less preferred by pigs once they are no long novel and that, therefore, the frequency of use decreases [19,20,21,22]. Trickett et al. found that the rope always generated less interaction when it was renewed than when it was first introduced [22]. However, in most publications, the concept of “renew” and “novel” are often mixed and difficult to be distinguished between.

This study had three aims. The first was to investigate whether pigs with undocked tails could be reared from birth to slaughter without tail biting when multiple enrichment items compatible with fully slatted floors were provided. Enrichments were selected based upon a variety of predetermined characteristics and types of presentation. Our second aim was to investigate the importance of novelty in enrichment provision. Pigs were divided into two treatment groups postweaning: SAME and SWITCH. SAME pigs had the same age-appropriate enrichment for the weaner and finisher stages, while the SWITCH pigs had enrichment materials changed every two weeks. The number of weeks each enrichment combination was provided was kept equal between the SAME and SWITCH treatments overall. Finally, the third aim was to determine whether pigs showed any preference for the different materials provided.

## 2. Materials and Methods

### 2.1. Ethical Statement

Ethical approval was obtained from the Teagasc Animal Ethics Committee (TAEC163–2017). All procedures were carried out in accordance with the Irish legislation (SI no. 543/2012) and the EU Directive 2010/63/EU for animal experiments.

The primary ethical concern for this study was that there would be a high risk of tail biting outbreaks due to not docking the pigs’ tails. To this end, pigs’ tails were inspected twice a day during the week and once daily during the weekend by one of the first two authors of the study (J.-Y. Chou or C. Drique), and routine checks were also performed by the technical staff in the research centre. A tail biting outbreak was defined as when 3 or more pigs in a pen were observed to have fresh, clearly visible blood present on their tails and/or when 3 or more pigs in a pen had tail damage reaching score 3 on the tail scoring system over a consecutive 72 hours. When an outbreak occurred, a set of predefined and randomised intervention protocols were deployed: 1. Removing victims, 2. removing suspected biters, or 3. adding in 3 ropes as additional enrichment and applying a layer of ointment (Cheno Unction, PharVet, Ireland) on all tails to reduce the smell of blood with a necessary treatment for the tail biting victims in place. In the case of pens with tail biting without meeting the outbreak criteria, the victims of severe biting were removed temporarily to a hospital pen for treatment for ethical reasons and reintroduced within a week. The hospital pens were located in the same room as experimental pens and had the same dimensions. As much as possible, pigs were accommodated with a partner from the same group to ease reintroduction after healing.

### 2.2. Animals and Housing

The experiment was carried out in the Pig Research Facility in Teagasc, Moorepark, in Ireland. Eight litters (Landrace × Large White) were used in the experiment, and all litters in the farrowing house were enriched with multiple enrichment materials. The piglets stayed in the farrowing pen (1.84 m × 2.50 m) for 4 weeks. One week after farrowing, a piece of hessian cloth (0.20 m × 0.20 m) and a small paper cup was provided, tied to the rear wall of the pen. A chewable plastic toy was put in at the beginning of the third week, and a piece of chopped bamboo (approximately 0.3 m in length, 0.05 m in diameter) at the beginning of the fourth week. The piglets were teeth-clipped, but their tails were kept undocked. No castration of the male piglets was performed. At weaning, the piglets were individually tagged and weighed, and 96 piglets were selected and randomly allocated to 8 treatment pens (12 pigs/pen) balanced for weight, sex (half male and half female), and litter mates (minimum 3 litter mates per pen). The mean weight between the pens and treatments was standardised as much as possible (minimum pen weight at 90.65 kg and maximum at 95.6 kg). The pigs were fed ad libitum with a standard pelleted diet through a single-spaced wet-dry feeding system, and water was also provided ad libitum via a nipple drinker. All pens were fully slatted with weaner pens dimensioned 2.4 m × 2.6 m and finisher pens 4.0 m × 2.4 m. The lighting was kept at 12 hours per day to ensure a normal circadian rhythm (around 150 lux for weaners and 130 lux for finishers). The temperature was thermostatically maintained at 27 °C immediately postweaning and dropped 2 degree every 2 weeks thereafter, controlled by automatic heating and automatically controlled mechanical ventilation. The pigs stayed in the weaner house for 7 weeks before being transferred to the finishing house without remixing. The finisher house temperature was maintained at 20 °C with the same automatically controlled mechanical ventilation.

### 2.3. Enrichment Treatments

Eight enrichment categories were created based on the different properties of enrichment items suitable for pigs, as outlined by Van De Weerd et al. (2003) [19] (Table 1):

Except the first category which was a common item present in all pens (an Easyfix^®^ floor toy in the weaner stage and a Piglyx^®^ lick block in the finisher stage; all items used in the experiment are listed in Table 2), four different enrichment items were identified, which fell into each category. All were easily and cheaply available in Ireland. The items within each category were then randomly assigned to one of four “combinations”. Thus, in each “combination”, there were 8 enrichment items (i.e., one from each of the 8 categories). To ensure the items could sustain wear and tear from use by larger pigs, more sturdy items were chosen for the finisher stage, but the characteristics of each category were consistent across stages.

Four pens of pigs were assigned to the “SAME” treatment. Each of these pens was assigned to a different one of the four “combinations” for the duration of the weaner and finisher stage. Four other pens of pigs were in the “SWITCH” treatment. For these pens, the “combination” they received was switched every two weeks. As such, the number of weeks each “combination” was provided was kept equal between the “SAME” and “SWITCH” treatments overall (see Appendix A for the detailed treatment assignment and distribution between pens and weeks).

During the experiment, the enrichment was monitored, cleaned, renewed, or replaced twice a day to ensure that pigs had access to all items at all times. The weight of the loose materials provided in a rack was recorded whenever it was renewed.

A previous study conducted on the same research farm by the authors showed that using a single enrichment item to rear undocked pigs under similar conditions led to a high occurrence of tail biting outbreaks with a large amount of pigs suffering from tail amputation [23]. Thus, due to ethical concerns and the legal requirement to provide pigs with manipulable materials, no negative control was used in the current experiment.

### 2.4. Pig Assessment

#### 2.4.1. Physical Measures

The pigs were weighed individually at weaning, upon moving to the finisher house, and before slaughter using weighing scales calibrated to the expected weight range of the pigs. Every 4 weeks, they were also weighed as a group to monitor the growth rate. The pen level feed intake was recorded through the computerized feeding system (BigFarmNet Manager, Big Dutchman Ltd. v3.1.5.51039, Vechta Calveslage, Germany).

Every two weeks, tail lesions, ear lesions, and tear staining levels were checked and scored on each pig individually by a single observer. The tail lesions were scored using the scoring system developed by the Farewelldock consortium [24]. One score was given according to the severity of the tail lesions observed (0: no lesion, 1: bite marks, 2: open wound, and 3: swollen bite wounds), and a second score was given based on the presence of blood (0: no blood, 1: black scar, 2: older red blood, and 3: fresh blood). Ear lesions (0: no lesion, 1: superficial scratches, 2: evidence of recent bleeding, 3: substantial cuts and bleeding, and 4: part of the ear missing) were scored with the same method detailed in Chou et al. [13]. Tear stains were recorded with the DeBoer–Marchant–Forde Scale (score 0–5), and both eyes were scored independently [25]. The pigs reached market age at 114 days postweaning. They were individually marked with slap marks the day prior to slaughter and traced through the factory to obtain postmortem data from the carcasses. The tails were scored on the slaughter line after scalding by a single trained observer, using a 0–4 scale modified from Harley et al. [26].

#### 2.4.2. Behaviour

• Direct behaviour observations

Observations of animal behaviour took place on two different days per week (except immediately postweaning and in the week after transferring to the finisher house) and twice per day: once in the morning around 10:30 h and once in the afternoon around 14:30 h, using a predefined ethogram (Table 3). These times were chosen based on a previous study [23] which indicated that this was when the pigs were most active. The pigs’ interactions with the enrichment items, their behaviours towards other pen mates, and play behaviour were the focus of the observations. Behaviour sampling was conducted continuously for 5 minutes in each session. All pigs were identified individually by a unique colour pattern marked on their back. The pigs were marked twice a week on the day before the observation day in order to minimise the disturbance.

• Video recording and behaviour scanning

All pens were video recorded (by QVIS HDAP400 CCTV cameras and Pioneer-16 digital recorder case, Hampshire, UK) continuously on one day every week, except for immediately postweaning and in the week after moving to the finisher house. Behavioural observations were carried out using scan sampling. A static image from the video recording was obtained every 20 minutes between 07:00 h and 20:00 h (*n* = 39 data points). The number of pigs interacting with each of the 8 enrichment items during that instant was counted. The identity or sex of the pigs was not recorded.

### 2.5. Statistical Analysis

The data were analysed using Statistical Analyses System (SAS, version 9.4, 1989, SAS Institute Inc., Cary, NC, USA). All residuals were checked for normality after analysis, and an appropriate transformation was applied (square root transformation) to non-normally distributed frequency data. For continuous and normal data such as the weight of the pigs, feed intake, the weight of loose materials, and behavioural frequency data, the MIXED procedure was used to fit mixed models. The stage (weaner/finisher), week within stage, treatment (SAME/SWITCH), and the enrichment combination within each stage were included as fixed effects in all behaviour analyses. When analysing the level of interaction with different enrichment items, category and the interaction between category and treatment were also included as additional fixed effects to investigate how often pigs interacted with each category. Direct behavioural observations were analysed as frequencies per pig per minute, and the interactions from the video scans were analysed as frequencies per pig per scan. For the analysis of the consumption of loose materials, week, treatment and the interaction between treatment and material were included. For the direct observations, sex was included as a fixed effect to investigate differences between male and female pigs. The behaviour data were averaged with week, which was included in the model as a repeated effect. All behavioural data were analysed using pen as the experimental unit (*n* = 8), except when comparing sex differences (*n* = 16). The growth data such as weight and feed intake were also analysed at the pen level (*n* = 8), as well as the weight of the loose materials (*n* = 8).

All physical scores (alive and postmortem) were analysed using the GLIMMIX procedure to fit the Generalised Linear Mixed Models, including week (or stage), treatment, and sex as fixed effects. This was analysed using individual pig as the experimental unit (*n* = 96).

## 3. Results

### 3.1. Consumption of Materials

Pigs in the SAME treatment consumed more grass than those in the SWITCH treatment in the weaner stage (SAME 104.9 ± 5.7 vs. SWITCH 28.6 ± 8.5 g/day, mean ± S.E., *p* < 0.001). Likewise, they consumed more grass (SAME 170.7 ± 7.6 vs. SWITCH 63.2 ± 10.8 g/day, mean ± S.E., *p* < 0.001) and miscanthus (SAME 234.7 ± 7.3 vs. SWITCH 130.4 ± 7.8 g/day, mean ± S.E., *p* < 0.001) in the finisher stage than the SWITCH pigs.

The replacement rate of other items only included the SAME groups since the items were always replaced in the SWITCH groups every 2 weeks according to the experimental design. Descriptive results on the frequency of replacement of the fabric materials (category 6) and the materials provided in the container (category 8) are provided in Table 4.

### 3.2. Physical Measures

Out of the 96 pigs used in the experiment, only one pig (from a SAME pen) received a severe tail biting injury during week 4. This pig experienced a removal of 2/3 of the tail and was removed from the experiment.

Although in both treatments tail lesion scores were low, there was a nonsignificant tendency for them to be numerically higher in the SAME pigs (Table 5). There was no difference in ear lesion or tear staining scores between the SAME and SWITCH pigs or in the postmortem tail lesion scores (Table 5). Neither was there an effect of sex on antemortem lesion scores or postmortem tail lesion scores. In terms of growth, the pigs in both treatments had the same average daily gain (ADG) and feed conversion ratio.

Female pigs had lower ADG than male (0.89 ± 0.01 vs. 0.97 ± 0.01 kg/day; F = 20.74_(85,1)_, *p* < 0.001) and lower live weights recorded before slaughter (female 107.58 ± 1.34 vs. male 116.45 ± 1.41 kg; F = 25.64_(85,1)_, *p* < 0.001).

The pigs had lower tail lesion scores in the weaner than in the finisher stage (0.72 ± 0.02 vs. 0.89 ± 0.02; mean ± S.E., F = 41.31_(658.1,1)_, *p* < 0.001), whereas ear lesion scores were worse in the weaner stage than later (1.02 ± 0.02 vs. 0.95 ± 0.02; mean ± S.E., F = 6.04_(667.1,1)_, *p* < 0.05). Figure 1 shows the pattern over time; tail lesion scores gradually increased from weaning until movement to the finishing stage, and then they stabilised (*p* < 0.001). Ear lesion scores were reasonably consistent across time, only lower in week 14 than in weeks 4 and 8 (*p* < 0.01, Figure 1).

### 3.3. Direct Behavioural Observations

There was no effect of the treatment (SAME or SWITCH) on any measure of behaviour.

#### 3.3.1. Enrichment Directed Behaviour

Table 6 summarises the amount of interactions with the enrichment overall and within each category in both the weaner and finisher stages. The pigs in the weaner stage were observed to interact more with enrichment in general, driven by more interaction with the common item (Easyfix for weaners and PigLyx for finishers) and the wood post. The only item for which there was less interaction in the weaner than finisher stages was the loose material in the container. There was no effect of sex on the amount of interaction with the enrichment.

#### 3.3.2. Preferences between and within Each Category

During the entire experiment, the pigs interacted more with the material in the rack than any other item and the least with the wood post and hanging wood (Figure 2, *p* < 0.001). Within each combination, a preference for certain items was also observed. Within the “floor toy” combination, weaners interacted more with the brush than with the dog toy (0.07 ± 0.01 vs. 0.01 ± 0.01; mean ± S.E., F = 3.82_(83.6,6)_, *p* < 0.01). In the finishing stage, the pigs were observed to interact most frequently with the cardboard roll among the fabric materials provided (cardboard = 0.07 ± 0.01, tonne bag = 0.04 ± 0.01, hessian sack = 0.04 ± 0.01, astroturf = 0.01 ± 0.01; mean ± S.E., F = 5_(56.7,6)_, *p* < 0.001), and they also interacted more with the sawdust than the dried grass provided in the hanging container (0.04 ± 0.01 vs. 0.01 ± 0.01; mean ± S.E., F = 2.71_(77.9,6)_, *p* < 0.05).

#### 3.3.3. General Pig Behaviour

A tail biting behaviour was performed more in the weaner (0.013 ± 0.002 instances/pig/min) than finisher (0.006 ± 0.001 instances/pig/min; F = 16.2_(177,1)_, *p* < 0.001) stages and by females (0.011 ± 0.001 instances/pig/min) more often than males (0.007 ± 0.001 instances/pig/min; F = 17.86_(7,1)_, *p* < 0.01). Likewise, ear biting was performed more in the weaner (0.025 ± 0.004 instances/pig/min) than finisher stages (0.020 ± 0.003 instances/pig/min; F = 5.2_(180,1)_, *p* < 0.05). Regarding bites to the other parts of the body (e.g., snout, legs, etc.), there was no effect of stage, but females performed more of this behaviour than males (0.0139 ± 0.013 vs. 0.0084 ± 0.013 instances/pig/min; F = 8.81_(14,1)_, *p* = 0.01). However, when all damaging behaviours were considered together (tail + ear + other body part directed), there was only a tendency for an effect of stage (weaner 0.060 ± 0.005, vs. finisher 0.047 ± 0.005 instances/pig/min; *p* = 0.06), and the effect of sex was just significant (Female = 0.063 ± 0.006, vs. male = 0.044 ± 0.006 instances/pig/min; F = 4.48_(14,1)_, *p* = 0.05).

Similar to damaging behaviours, pigs performed more positive behaviours (sum of social nosing and play) in the weaner than finisher stages (0.031 ± 0.003 vs. 0.022 ± 0.002 instances/pig/min; F = 10.09_(180,1)_, *p* < 0.01), and females performed more than males (0.029 ± 0.002 vs. 0.023 ± 0.002 instances/pig/min, respectively, F = 7.93_(14,1)_, *p* = 0.01). Aggression was also higher in the weaner stages than finisher (0.103 ± 0.007 vs. 0.054 ± 0.005 instances/pig/min, respectively; F = 50.42_(180,1)_, *p* < 0.001), yet contrary to all other behaviours, males performed more aggression than females (0.088 ± 0.006 vs. 0.069 ± 0.006 instances/pig/min; F = 5_(14,1)_, *p* < 0.05).

### 3.4. Video Scans

Unlike the direct observations, the SAME pigs were recorded interacting with loose materials in the rack more frequently than the SWITCH pigs (0.20 ± 0.01 vs. 0.13 ± 0.01, mean ± S.E., *p* < 0.001) by video scans.

Similar to the results from the direct observations, the video scans also revealed that weaner pigs interacted more with the enrichment than finishers (Table 6). In this case, it was driven by an increased level of interaction with all items other than the hanging wood (no difference) and the loose material in a container, which, again, was interacted with less in the weaner than finisher stages. Over the entire experiment, similar to the direct observations, the rack of loose material received the most interactions, and the wood received the least (Figure 2; *p* < 0.001).

Between different types of floor toy, weaners interacted more with the brush than with the dog toy (0.020 ± 0.003 vs. 0.004 ± 0.003; mean ± S.E., F = 4.01_(53.7,6)_, *p* < 0.01) and more with the Easyfix floor toy than the dog toy (0.018 ± 0.003 vs. 0.004 ± 0.003; mean ± S.E., p < 0.05). In the “hanging chew toy” category, pigs in the weaner stage interacted more with bamboo pieces than with tennis and the dog tug toy (bamboo = 0.010 ± 0.002, tennis = 0.003 ± 0.002, dog toy = 0.003 ± 0.002; mean ± S.E., F = 3.33_(85.6,6)_, *p* < 0.05), and in the “container” category, they interacted more with peat than with the ground sawdust and grated carrot (peat = 0.0070 ± 0.0013, ground sawdust = 0.0003 ± 0.0013, dog toy = 0.0010 ± 0.0013; mean ± S.E., F = 2.94_(57.6,6)_, *p* < 0.05).

## 4. Discussion

This study investigated the possibility of successfully rearing pigs without tail docking by using multiple slat-compatible enrichment items; we explored the effect of variation in enrichment provision and further compared pigs’ preferences for different enrichment items. The results showed that, although a low level of tail biting occurred, the lesions were mild (lesion scores around 1 or less on a 0–3 scale) and that there were no tail biting outbreaks. Overall, providing pigs with a more varied enrichment had a marginally positive effect on the severity of tail lesions, whereby pigs given more variation tended to have lower tail lesion scores, although this was not statistically significant. Pigs also showed clear preferences for certain enrichment items, with the loose materials in a rack being most preferred.

### 4.1. Overall Effect of Enrichment On Tail Biting

Throughout the experiment, only one pig was severely tail bitten, with 2/3 of its tail amputated. No tail biting incidents reached the outbreak severity criteria set out beforehand. However, tail biting behaviours were still observed, and tail lesions scores became progressively worse over time, which is a characteristic of tail biting behaviour. The increase in tail lesion scores is more likely due to the low levels of biting on an ongoing basis, which were not acutely damaging to the recipients. The results demonstrated that it was possible to keep tail biting at a manageable level even among intact tail pigs on fully slatted floors by a high standard of enrichment provision and management that was compatible with the housing system.

### 4.2. Differences between SAME and SWITCH Groups

There was no difference in the general behaviour of the pigs that received a consistent type of enrichment (SAME) and pigs that had their enrichment types switched regularly (SWITCH) even though the SAME pigs had a slight but not statistically significant tendency to have worse tail lesions. Scott et al. suggested that it is the “stimulus properties” of the enrichment rather than the quantity alone that is important to sustain the interest of pigs [29]. In the current study, the SWITCH pigs could be more stimulated by the change in enrichment and, therefore, did not tail bite as much as the SAME pigs.

Although the level of interaction with the enrichment did not differ between the SAME and SWITCH pigs when observed directly, the results from the video observations showed that the SAME pigs interacted with the loose materials in the rack more than the SWITCH pigs. Moreover, when provided with grass, the SAME pigs consumed more than the SWITCH pigs. The loose materials were replenished whenever the rack was emptied, and a higher replenishment rate could have increased the novelty effect for the SAME pigs, which in turn stimulated more use of the rack. Van de Weerd et al. found that the trait “renewed” was desirable for pigs and generated more interaction [19]. Trickett et al. also demonstrated that, whenever a point source enrichment item was replenished and reintroduced, it regenerated a higher frequency of interaction than a pre-existing item [22]. Thus, the attractiveness of the item and the effect of novelty might have an additive effect on the level of interaction. As the SWITCH pigs were exposed to multiple new enrichment items every two weeks, the novelty effect of all the items present in the pen may have diverted their attention away from the loose materials in the rack, even though, overall, they were still the most preferred by the pigs. The reason why the difference was only picked up by video scanning could be due to the observations including the whole day, rather than focused on the most active times of day.

### 4.3. Pigs’ Preference Forenrichment Categories and Items

Regardless of age or treatment, pigs interacted with the loose materials in the rack most frequently; this is in agreement with previous research showing that loose straw or straw in racks occupies pigs’ time more than other items [29,30,31,32]. Nevertheless, there have been some studies where pigs interacted with point-source enrichment (a straw pellet dispenser) more than loose materials (chopped straw/miscanthus) provided as floor litter [33]. Guy et al. also found that pigs preferred rope and chain to sawdust presented in a trough [34]. In the current experiment, there was no available litter on the floor. Thus, the material in the rack was likely the next best substitute and the most appealing for the pigs regardless of which type of materials used. The elevated rack we used kept the materials clean and easily accessible (up to 6 pigs could use it simultaneously).

Second to the loose materials in the rack, the floor toys and fabrics were the categories that pigs interacted with most. The floor toys were moveable, which increased the accessibility compared with the items that were installed at a fixed location. Scott et al. also mentioned social facilitation, whereby pigs were stimulated by pen mates, and therefore, synchronising their behaviour could increase pigs’ interaction with enrichment [29]. The fabric, on the other hand, was soft and destructible, which were properties known to be favourable to pigs [19]. In contrast, both the wooden post and hanging wood attracted the least attention, which agreed with a previous study which compared wood blocks with ropes [22]. Telkänranta et al. found that pigs preferred fresh wooden pieces to chains but no difference between wood and polythene pipes [35]. The wood used in the current trial was not freshly cut but commercially dried, which could have reduced the pigs’ interest.

The bamboo piece (compared with tennis ball, rubber pipe, and dog tug toy) and cardboard roll (compared with cotton piece, coconut basket, and hessian cloth), which were preferred by the pigs, both possess relatively more attractive characteristics compared to other items in their categories. Van de Weerd et al. identified the main characteristics of objects which captured and sustained the interest of pigs to be ingestible, odorous, chewable, deformable, and destructible [19]. A study also found that pigs interacted more with softer wood species than the hard [13]. In the current experiment, items that were more easily destroyed (destructible) also generated a higher amount of interaction and needed to be replaced more often. Another factor influencing the use of enrichment is cleanliness. Items that are easily soiled by excretion were used less by pigs [30]. Based on the experimenter’s experience, the dog toy was constantly soiled and needed to be cleaned compared with the brush, rubber boot, and Easyfix^®^ floor toy, which may have contributed to its low rate of use. It was smaller and lighter than the other items, so was more often covered in faeces when presented on the floor, which could have inhibited its use by the pigs.

The location of enrichment, particularly with regard to point source items which are fixed in a specific location in the pen, could affect the access and competition [30]. In the current study, the loose materials provided in the container were presented differently in the weaner and finisher stages: an open-top bucket hung at the side of the gate near the feeder for the weaners and a sealed canister suspended in the centre of the pen was used for the finishers. The finishers interacted with this category more than the weaners. This is consistent with the suggestion by Van de Weerd et al. [30] that, even when all other properties of the enrichment were the same, a central location could invite easier access and, therefore, more interaction than an item placed peripherally. Thus, factors such as location in the pen and presentation method need to be considered carefully when choosing environmental enrichment options for pigs.

### 4.4. Effects of Age and Sex

Weaners performed both more positive and harmful behaviours than finishers and interacted more with the enrichment, which could be due to pigs being more active in general when they are younger. Indeed, the video observations showed that weaners had more frequent interactions with most categories of enrichment items than finishers, except for the hanging wood. Docking et al. also found an effect of age of the pig on the level of interaction with objects, with finishers showing less interaction with objects than weaners [36]. Other studies have also shown that ear biting behaviours decreased as time progressed in the later weaner stage [37].

Nevertheless, tail lesion scores of finishers were marginally higher than those of weaners, which again agrees with previous studies [38]. This could be due to the accumulation of tail biting damage over time.

In the current study, female pigs performed more biting behaviour in general than males. As the females were lighter and had a lower average daily gain than males, they perhaps used biting behaviours to gain access to enrichment or the feeder [14]. Although some studies have shown that male pigs tended to have more tail lesions than females [39], others have shown no difference [40], similar to our results. In terms of aggressive behaviours, male pigs generally perform more aggression than females [41], and this could be particularly the case when males are not castrated.

## 5. Conclusions

This experiment demonstrated that it is possible to rear undocked pigs on fully slatted floors when a relatively large variety, quantity, and quality of slat-compatible enrichment is provided. Although mostly minor tail biting events occurred, the enrichment provided sustained pigs’ interest and limited the escalation of tail biting towards an outbreak. This underlines the importance of environmental enrichment in keeping pigs sufficiently stimulated and in reducing harmful behaviours. Varying the types of enrichment provided over time did not significantly reduce tail damage. Similarly, no differences in damaging behaviours were observed, but pigs having the same enrichment interacted with the loose materials in the rack more. The study also showed that pigs have preferences for certain enrichment materials, with loose materials in a long rack being used the most, and therefore, it is important to consider enrichment characteristics, presentation, location, and maintenance when providing enrichment. The next step is to find a balance between the level of environmental enrichment that needs to be provided to reduce the risk of tail biting and the level that is economically feasible and practical to manage for farmers.

## Figures and Tables

**Figure 1 animals-09-00139-f001:**
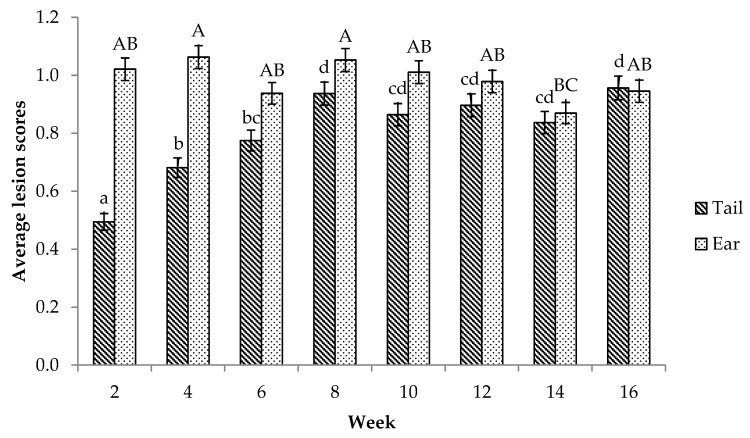
The mean ± S.E. tail and ear lesion scores across time (n = 48 per treatment group): The tail lesion score was the average of the 2 tail scores given, where one score represented the severity of the lesions (0: no lesion, 1: bite marks, 2: open wound, and 3: swollen bite wounds), and the other represented the presence of blood (0: no blood, 1: black scar, 2: older red blood, and 3: fresh blood). Ear lesions were scored on a 0–4 scale (0: no lesion, 1: superficial scratches, 2: evidence of recent bleeding, 3: substantial cuts and bleeding, and 4: part of the ear missing).

**Figure 2 animals-09-00139-f002:**
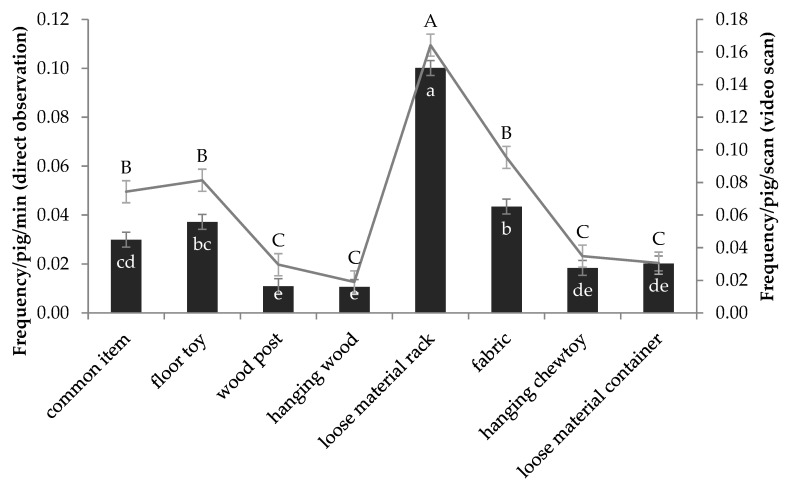
The overall pig interaction frequency with each category of enrichment at a pen level (n = 8) via direct observation (bar chart, different small letters denote significant difference) and video scans (line chart, different capital letters denote significant difference). The data are presented as the mean ± S.E.

**Table 1 animals-09-00139-t001:** The enrichment categories used in the experiment.

Properties [19]						
Categories	Rootability	Durability	Edibility	Presentation	Texture	Location
1. Common item *	-	-	-	-	-	-
2. Floor toy	Yes	Deformable	Chewable	Movable	Soft	Floor
3. Wood post	Yes	Destructible	Edible	Attached	Hard	Floor
4. Hanging wood	No	Destructible	Edible	Suspended	Hard	Eye level
5. Loose material (long rack)	No	Renewed	Edible	Attached	Loose	Eye level
6. Fabric	No	Destructible	Chewable	Suspended	Soft	Eye level
7. Hanging chewtoy	No	Deformable	Chewable	Suspended	Soft	Eye level
8. Loose material (container)	No	Renewed	Edible	Suspended	Loose	Eye level

* The same item was present in all pens (an Easyfix^®^ floor toy for weaners and a Piglyx^®^ lick block for finishers).

**Table 2 animals-09-00139-t002:** The enrichment item combinations from different categories in randomised groups.

Stage	Weaner				Finisher			
Combinations/Categories	1	2	3	4	1	2	3	4
1. Common item	Easyfix^®^ floor toy	Easyfix^®^ floor toy	Easyfix^®^ floor toy	Easyfix^®^ floor toy	PigLyx^®^	PigLyx^®^	PigLyx^®^	PigLyx^®^
2. Floor toy	Brush	Dog toy	Rubber boot	Easyfix floor toy	Larch	Easyfix floor toy	Cherry	Spruce
3. Wood post	Larch (M)	Pine	Spruce	Larch (S)	Spruce	Pine	Larch (S)	Larch (M)
4. Hanging wood	Pine	Bamboo	Larch	Spruce	Pine	Larch	Spruce	Bamboo
5. Loose material (long rack)	Grass	Straw	Sawdust	Shredded paper	Grass	Straw	Miscanthus	Shredded paper
6. Fabric	Cotton piece	Cardboard roll	Coconut basket	Hessian cloth	Cardboard roll	Tonne bag	Hessian sack	Astroturf
7. Hanging chewtoy	Tennis balls	Rubber pipes	Dog tug toys	Bamboo pieces	Rubber boot	Dog toy frisbee	Bamboo stick	Dog tug toy
8. Loose material (container)	Sawdust powder	Shredded grass	Peat	Chopped carrots	Shredded paper (S)	Sawdust powder	Peat	Dried grass

**Table 3 animals-09-00139-t003:** The ethogram used for behaviour observation.

Enrichment Directed Behaviours	Description of Behaviours
Bite device	Oral manipulation of the device with mouth open
Root device	Manipulation of the device by manoeuvring the device locomotively using the snout
Aggressive encounter	Biting, headknocking, or pushing over access to the device
Other	Any physical contact with the device other than mouth/snout
**Negative behaviours**	
Tail manipulation not at feeder (standing or lying/sitting)	Oral manipulation of the tail of another pig away from the feeder.
Tail manipulation at feeder	Oral manipulation of the tail of another pig which is feeding in the vicinity of the feeder.
Ear manipulation (standing or lying/sitting)	Oral manipulation of the ear of another pig.
Biting other parts of the body	Biting pen mate in another region other than tail and ear, e.g., hock, flank, snout, or genital area
Belly nosing	Rubbing/manipulating a pen mate’s belly/flank region with rhythmic up and down snout movement
Mounting	Putting two front legs on top of another pig
Aggressive behaviour	Pushing, headknocking, and open-mouth fighting with pen mates
**Positive behaviours**	
Social nosing (face)	Gentle, non-open mouth nosing on another pen mate’s facial area (without aversive reaction from the recipient)
Individual play	Based on Newberry et al. (1988) [27] and Donaldson et al. (2002) [28], any scampering, pivoting, head tossing, flopping, and pawing movement.

**Table 4 animals-09-00139-t004:** The descriptive replacement rate for the fabric and loose material provided in the container in two different stages among the SAME groups.

Stage	Weaner				Finisher			
Category/Group	Fabric	Days (mean ± S.E.)	Container	Days (mean ± S.E.)	Fabric	Days (mean ± S.E.)	Container	Days (mean ± S.E.)
1	Cotton	16.3 ± 14.3	Sawdust	2.2 ± 0.4	Cardboard	2.4 ± 0.5	Paper	62.0 *
2	Cardboard	4.9 ± 1.8	Grass	2.6 ± 0.4	Tonnebag	64.0 *	Sawdust	8.9 ± 4.3
3	Coconut	5.4 ± 1.3	Peat	1.2 ± 0.1	Hessian	10.7 ± 3.6	Peat	8.9 ± 3.4
4	Hessian	9.8 ± 3.3	Carrot	1.8 ± 0.2	Astroturf	8.0 ± 4.1	Dried Grass	62.0 *

* The items were not replaced during the duration of the trial, and therefore, no S.E. is available.

**Table 5 animals-09-00139-t005:** The lesion scores and tear stains for the SAME and SWITCH groups: The score data are presented as mean ± S.E. (n = 48 per treatment group).

Scores	SAME	SWITCH	F-Value	*p*-Value
Tail (alive)	0.83 ± 0.03	0.76 ± 0.03	3.43	0.07
Ear	0.98 ± 0.02	0.98 ± 0.02	0.01	>0.05
Tear staining	1.83 ± 0.07	1.79 ± 0.07	0.2	>0.05
Tail (postmortem)	1.00 ± 0.15	0.80 ± 0.13	1.01	>0.05

**Table 6 animals-09-00139-t006:** The pig interaction with different enrichment categories during the weaner and finisher stages via direct observation and video scans: The data presented are frequencies per pig per minute at a pen level, as the mean ± S.E. (*n* = 8).

Categories	Direct				Video			
	Weaner	Finisher	F-Value	*p*-Value	Weaner	Finisher	F-Value	*p*-Value
All items	0.288 ± 0.012	0.253 ± 0.011	5.34	<0.05	0.083 ± 0.003	0.053 ± 0.002	75.52	<0.001
Common item	0.043 ± 0.003	0.019 ± 0.003	29.26	<0.001	0.014 ± 0.001	0.006 ± 0.001	44.98	<0.001
Floor toy	0.039 ± 0.006	0.035 ± 0.005	0.42	>0.05	0.013 ± 0.001	0.008 ± 0.001	10.77	<0.01
Wood post	0.016 ± 0.002	0.007 ± 0.002	9.17	<0.01	0.006 ± 0.001	0.002 ± 0.001	27.01	<0.001
Hanging wood	0.012 ± 0.002	0.009 ± 0.002	1.51	>0.05	0.002 ± 0.000	0.002 ± 0.000	0.01	>0.05
Loose material (long rack)	0.101 ± 0.011	0.096 ± 0.010	0.33	>0.05	0.024 ± 0.002	0.019 ± 0.002	3.98	<0.05
Fabric	0.039 ± 0.007	0.048 ± 0.006	1.35	>0.05	0.016 ± 0.002	0.009 ± 0.002	8.23	<0.01
Hanging chewtoy	0.023 ± 0.003	0.014 ± 0.002	5.63	<0.05	0.006 ± 0.001	0.003 ± 0.001	9.95	<0.01
Loose material (container)	0.015 ± 0.004	0.024 ± 0.003	6.68	<0.05	0.002 ± 0.001	0.005 ± 0.001	8.83	<0.01

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
