# Peer review of "Rearing Undocked Pigs on Fully Slatted Floors Using Multiple Types and Variations of Enrichment"

_animals, 2019, doi:10.3390/ani9040139_

Round 1

Reviewer 1 Report

Manuscript review

Thank you for the opportunity to review the manuscript by Chou et al, looking at enrichment in undocked pigs. This is a comprehensive study that is well designed and analysed in my opinion. The variation in enrichment strategy and types of enrichment are extensive and provide evidence that will prove useful in determining a viable strategy for commercial situations.

I have a very small number of minor comments below. I have had some difficulties with the title elements of “multiple types” and “variety of” enrichment – I cannot decide if variety implies multiple types, whereas we are seeing “variation in enrichment regime”.  If the editor and reviewers are fine with the title then please ignore as I may be over thinking it. The current title adds brevity where I think I may tend to overcomplicate.

Minor comments:

Line 145 (Table 1) – Rootability column has YES / NO and I’m not sure the CAPS sit well against the other columns (personal opinion here rather than fundamental issue)

Line 189 – “chosen times based on previous study” – do we need a reference here or “unpublished study”?

Line 505 and 508 – incomplete refs or a different form of reference to published papers?

Author Response

Response to Reviewer 1 Comments

Point 1: I have a very small number of minor comments below. I have had some difficulties with the title elements of “multiple types” and “variety of” enrichment – I cannot decide if variety implies multiple types, whereas we are seeing “variation in enrichment regime”.  If the editor and reviewers are fine with the title then please ignore as I may be over thinking it. The current title adds brevity where I think I may tend to overcomplicate.

Response 1: We agreed with the reviewer’s suggestions that “variety” can be more ambiguous compared to “variation.” Line 3, 27, 337, 342 amended.

Point 2: Line 145 (Table 1) – Rootability column has YES / NO and I’m not sure the CAPS sit well against the other columns (personal opinion here rather than fundamental issue)

Response 2: Line 146 (Table 1) amended accordingly.

Point 3: Line 189 – “chosen times based on previous study” – do we need a reference here or “unpublished study”?

Response 3: Line 195 amended (reference 23 added in, line 531-533).

Point 4: Line 505 and 508 – incomplete refs or a different form of reference to published papers?

Response 4: Line 512-513, 516-517 amended.

Reviewer 2 Report

This is a good paper and the content is important. However, I cannot over look the fact that there is no data to indicate what the tail biting rates are without enrichment. For this reason I don't think that the first aim of the paper has been met; it does address whether pigs can be reared from birth to slaughter without tail biting BUT there is no way to determine if this has anything to do with the enrichment. This could be rectified by doing a farm survey of the tail biting rates on this particular farm when they do not tail dock and do not use enrichment or by running a small batch of both the enrichment groups and also a control group where pigs have tails and no enrichment. Alternatively you could change the angle of the article to merely compare SAME and SWITCH however I don't think this would have as much impact. At this stage there is no proof in this paper that enrichment decreases tail biting, only that tail biting is altered by switching enrichment types.

However, it was very well written and do I encourage you to try and address this issue as the article shows great promise.

Author Response

Response to Reviewer 2 Comments

Point 1: This is a good paper and the content is important. However, I cannot over look the fact that there is no data to indicate what the tail biting rates are without enrichment. For this reason I don't think that the first aim of the paper has been met; it does address whether pigs can be reared from birth to slaughter without tail biting BUT there is no way to determine if this has anything to do with the enrichment. This could be rectified by doing a farm survey of the tail biting rates on this particular farm when they do not tail dock and do not use enrichment or by running a small batch of both the enrichment groups and also a control group where pigs have tails and no enrichment. Alternatively you could change the angle of the article to merely compare SAME and SWITCH however I don't think this would have as much impact. At this stage there is no proof in this paper that enrichment decreases tail biting, only that tail biting is altered by switching enrichment types.

Response 1: We’d like to thank the reviewer for reminding us of the absence of the negative control group. The reasoning behind this is the fact that before the current study, there was indeed an experiment conducted by the same authors on the same farm, with identical husbandry procedures to rear undocked pigs, other than a much lower amount of enrichment/pig. In that experiment (n=672) only a single enrichment item was provided per pen of pigs and it led to serious tail biting incidents: 26 occurrences of tail biting outbreaks (with the same definitions used in the current study), 197 pigs suffered from severe tail damage that needed to be removed for treatment, 58 pigs removed as tail biters, 50 pigs were treated with antibiotics in home pens, and 11 pigs were permanently removed from the experiment due to tail biting. Based on this experience and the concern conveyed by our ethics committee, we decided not to use a negative control, but focus on a good/better provision of enrichment. That experiment is currently being prepared for publication and therefore we added in this explanation in line 167-171 and cited our manuscript in preparation (reference 23). As mentioned in our introduction line 76-79, other researchers are still struggling finding ways to rear undocked pigs on fully-slatted floors with manageable level of tail biting. As tail docking is still considered a necessity on most EU farms, our focus is to demonstrate that by using good quantity and quality of enrichment that is compatible with fully-slatted floor, it is possible to rear undocked pigs in the current system.

Round 2

Reviewer 2 Report

Thank you for clarifying the base levels of tail biting on this farm.